# Skill, or Style? Classification of Fetal Sonography Eye-Tracking Data

**Clare Teng**                                                CLARE.TENG@ENG.OX.AC.UK
*Institute of Biomedical Engineering, University of Oxford, United Kingdom*

**Lior Drukker**                                                DRUKKER@GMAIL.COM
*Nuffield Department of Women's and Reproductive Health, University of Oxford, United Kingdom*
*Women's Ultrasound, Department of Obstetrics and Gynecology, Beilinson Medical Center, Sackler*
*Faculty of Medicine, Tel-Aviv University, Israel*

**Aris T. Papageorghiou**                          ARIS.PAPAGEORGHIOU@WRH.OX.AC.UK
*Nuffield Department of Women's and Reproductive Health, University of Oxford, United Kingdom*

**J. Alison Noble**                                        ALISON.NOBLE@ENG.OX.AC.UK
*Institute of Biomedical Engineering, University of Oxford, United Kingdom*

## Abstract

We present a method for classifying human skill at fetal ultrasound scanning from eye-tracking and pupillary data of sonographers. Human skill characterization for this clinical task typically creates groupings of clinician skills such as expert and beginner based on the number of years of professional experience; experts typically have more than 10 years and beginners between 0-5 years. In some cases, they also include trainees who are not yet fully-qualified professionals. Prior work has considered eye movements that necessitates separating eye-tracking data into eye movements, such as fixations and saccades. Our method does not use prior assumptions about the relationship between years of experience and does not require the separation of eye-tracking data. Our best performing skill classification model achieves an F1 score of 98% and 70% for expert and trainee classes respectively. We also show that years of experience as a direct measure of skill, is significantly correlated to the expertise of a sonographer.

**Keywords:** Eye-tracking, skill classification, fetal ultrasound

## 1. Introduction

The definition of human skill in the medical literature is most often quantified by the number of years of experience a trained medical professional has been practicing for. In fetal sonography (pregnancy ultrasound screening), this corresponds to the number of years after qualification. In Wang et al. (2020); Sharma et al. (2021c), a sonographer who has been scanning for 2 years or less is defined as newly qualified. In Lous et al. (2021), a trained professional who has been scanning for 10 years or more is considered an expert. In other clinical sub-specialties such as surgery, skill is referenced to the number of instances the specific surgery has been performed (Ortega-Morán et al., 2020; Erridge et al., 2018). Similarly, in dentistry, the number of semesters completed by a trainee is used as a measure of skill (Castner et al., 2018, 2022). These time-based definitions are over simplified and

omit other important factors that contribute to skill level. Some examples are the frequency of scanning over time, quality (Wang et al., 2022) and interpretation of the recorded image and real-time response to visual feedback (Drukker et al., 2021). Since maternal and fetal anatomy differs, no two patients will present in the same manner at any given time (Drukker et al., 2021). These measures of skill are not easily quantifiable, and current definitions used for skill groupings are domain specific.

In medical studies where eye-trackers have been used for skill assessment, researchers typically use metrics such as the number of fixations and saccades, and the time taken to complete the task to differentiate groups of clinicians (Topalli and Cagiltay, 2018; Castner et al., 2022; Fichtel et al., 2019; Law et al., 2004). For example, Lee and Chenkin (2021) showed that experts spent significantly less time fixating on a relevant area-of-interest and had a higher fixation count compared to trainees when viewing video clips of an ultrasound examination. These studies depend on suitable experts or eye movement classification algorithms to separate eye-tracking data into fixations, saccades, smooth pursuits, and areas-of-interest.

Separating eye-tracking data into different eye movements is challenging. Research has shown that the selected eye movement classification algorithm is heavily dependent on the chosen parameters, and can return vastly different results within in the same domain-specific application (Salvucci and Goldberg, 2000). In fetal sonography, separating task-specific eye tracking into eye movements is made more challenging because of the number of diagnostic planes that need to be captured and assessed; capturing and reading each anatomical plane is considered a separate task. In second-trimester scanning specifically, there are 23 planes to be captured in a 30-40 minutes appointment window (Drukker et al., 2021).

A second question is how to define human skill for this task. In surgery, for instance, it has been proposed to measure skill by the time taken to complete the task and whether each suture was closed correctly (Reiley and Hager, 2009). In radiology, how far the radiographer deviated from the problematic areas could be an indicator that they did not identify the lesion as quickly as another expert (Manning et al., 2006; Krupinski et al., 2014). These definitions do not account for the nuances in fetal sonography, where skill metrics based on eye or probe/hand motion is non-trivial, due to the fast probe movement, fetal movement, unstructured transitions between numerous anatomical planes, sonographer experience, and maternal and fetal anatomy (Drukker et al., 2021). We test the hypothesis of whether grouping sonographer expertise based on years of experience is a suitable measure of skill.

## 1.1. Related Work

There are several studies that use eye-tracking data for task-specific skill classification. However, it is more typical to use tool motion data either on its own or in combination with other data modalities in applications such as surgery and fetal sonography (Lin et al., 2006; Megali et al., 2006; Wang et al., 2020), as opposed to only eye-tracking data. A Hidden Markov model is used in Ahmidi et al. (2010) to classify skill between experts and novices using both their eye-tracking and tool motion data. A statistical model was fitted to eye-tracking and tool motion data for experts and novices in endoscopic sinus surgery (Ahmidi et al., 2012). In the field of fetal sonography, Sharma et al. (2021b) uses a combination of eye-tracking, pupillary data, and image data to classify newly qualified

and expert sonographers. To define skill, Ahmidi et al. (2010) uses anatomical knowledge and operational knowledge of endoscopes as skill indicators. However, in fetal ultrasound, prior skill characterisation studies that use eye tracking or probe motion have used years of experience as a skill indicator (Sharma et al., 2021c,b; Wang et al., 2020; Lous et al., 2021). These studies combine eye-tracking with other data modalities, did not consider task-agnostic gaze behaviour, and are still largely limited by a year threshold cut-off to define skill (Ahmidi et al., 2010, 2012; Wang et al., 2020, 2022; Sharma et al., 2021c).

### 1.2. Contribution

Our main contributions are as follows. We build a task-agnostic skill classification model using only eye-tracking and pupillary data of sonographers performing fetal ultrasound scans. We calculate the correlation between years of scanning experience and the proportion of predicted expert labels of fully qualified sonographers and its significance (at the 5% level). To determine how well a task-agnostic skill classification model performs on specific tasks, we calculate the significance for anatomical planes with different levels of difficulty.

## 2. Method

We present an original skill classification model to differentiate trainee and fully-qualified sonographers based on their eye gaze characteristics, and use this to evaluate if years of scanning experience is an indicative measure of skill.

### 2.1. Skill Classification Model

An *expert* refers to any fully-qualified (FQ) sonographer, independent of their years of scanning experience. A *trainee* refers to a not yet fully-qualified sonographer, who is still learning how to scan. A *teacher* is a fully qualified sonographer who is performing the scan with a trainee present. We define *style* as in Wang et al. (2020); the gaze of a sonographer is the outcome of both human skill, and personal scanning style. The purposes of the skill classification model are 2-fold. The first is to differentiate a trainee and an expert's gaze behaviour using eye-tracking and pupillary data. The second is to determine if a sonographer's style of scanning affects the performance of the model.

To achieve these aims, we train several groups of models using different experts in the training dataset.

*Group 1: Teacher VS trainee* We aim to differentiate gaze of a teacher (FQ sonographer) and a trainee. We compare the differences between using a sonographer, as 1) the Teacher and 2) the same sonographer carrying out scans individually. During the training sessions, due to time constraints, the trainee does not necessarily perform the scan but is instead given opportunities to try searching for planes with some guidance from the teacher.

*Group 2: FQ sonographers VS trainee* This group of models aim to differentiate a population of FQ sonographers from a trainee. Here, FQ sonographers are performing scans individually and in some instances, with a trainee.

*Group 3: FQ sonographer VS trainee* The final group of models aim to differentiate a single FQ sonographer from a trainee. This is a reversed leave-one-out approach, analogous

to 'leave-one-in'. The null hypothesis is that each individual sonographer is able to provide a representation of the gaze behaviour of all expert sonographers.

We use eye-tracking data collected when sonographers were viewing live B-mode ultrasound video streams. Live B-mode video streams are recorded when the sonographer is actively searching for the required anatomical plane. In contrast, frozen frames result when the sonographer has frozen the video and is no longer moving the probe. A fetal ultrasound video typically follows an alternating sequence of live B-mode streams and frozen streams that are referred to as live B-mode and frozen *segments*, respectively. Only live B-mode segments are used for this study.

Instead of labeling the video in terms of the anatomy being assessed at this time as in Sharma et al. (2021a); Wang et al. (2020, 2022), we consider a task-agnostic approach to classify skill differences. This reduces the need for manual labelling of segments and builds a gaze-based skill classification model that is agnostic to the type of anatomical plane being searched for. To overcome the problem that live B-mode segments are of different lengths, we extract summarized gaze characteristics for each segment using the scalable feature extraction approach `tsfresh` (Christ et al., 2018).

**Gaze Features**  Following Sharma et al. (2021b) where pupillary data was used to compare differences between sonographers with $> 2$ years and $\leq 2$ years of experience, we calculated the task-evoked pupillary response (TEPR) as a skill classification feature. Briefly, TEPR measures the change in pupil dilation with respect to a baseline pupil diameter. A larger change in TEPR is indicative of a higher cognitive load, and vice versa. The equation for calculating TEPR is given as $\delta d_t$ in Equation (1). Following Sharma et al. (2021b), we use the minimum pupil diameter $d_r$ to represent the sonographer's pupil diameter while resting. $d_t$ represents the pupil diameter at time $t$ and $\delta d_t$ represents the TEPR at time $t$.

$$\delta d_t = \frac{d_t - d_r}{d_r} \times 100\% \tag{1}$$

We also include gaze data (x and y coordinates) as features. Each live B-mode segment is represented by a $3 \times n$ feature vector, where $n$ is its segment length and 3 is the number of final features that were used to train the model - gaze x and y co-ordinates and $\delta d_t$. Note that $n$ varies from segment to segment. The feature is then reduced to a $1 \times m$ feature vector, where $m$ is the number of characteristics extracted using Christ et al. (2018).

The feature extraction setting used in `tsfresh` was `EfficientParameters` (Christ et al., 2018) which consist of 74 unique time-series features. This setting was chosen because they provide an overview of time-series properties that are not computationally expensive to calculate and is scalable for large datasets. These features cover a range of time-series properties, such as distribution of data points, correlation properties, stationarity, entropy, and nonlinear time series analysis (Christ et al., 2018). In fetal ultrasound, due to the unstructured nature of searching for anatomical planes, the time taken per segment is not necessarily a fair indicator of skill. Hence we remove features related to length: `length`, and `ratio_value_number_to_time_series_length`, which calculate the length of the segment $n$ and the number of unique values in the segment divided by $n$, respectively.

The final dataset consists of a matrix of size $d \times m$, where d is the total number of segments available.

**Implementation** We consider off-the-shelf gradient boosting decision trees which have been shown to have the best performance on tabular data (Bentéjac et al., 2021) and are computationally efficient. These models are Categorical Boosting (CatBoost) (Dorogush et al., 2018), Light Gradient Boosting Machine (LightGBM) (Ke et al., 2017), and Extreme Gradient Boosting (XGBoost) (Chen and Guestrin, 2016) classifiers. Gradient boosting decision trees use an ensemble of weak decision trees to build strong predictors (Dorogush et al., 2018; Friedman, 2001). Briefly, XGBoost is a highly scalable and efficient gradient tree boosting algorithm that can handle sparse tabular data because of its algorithmic optimisations detailed in Chen and Guestrin (2016). LightGBM included two extra optimisation steps to handle large amounts of data instances and features, decreasing the computational speed and memory required compared to XGBoost (Ke et al., 2017). CatBoost is similar to XGBoost and LightGBM but is specifically designed to handle categorical features (Dorogush et al., 2018), of which there are several in the extracted features from `tsfresh`.

We performed a 5-fold stratified cross-validation with 75% of our dataset, and tested on the remaining 20%. We tuned our model parameters using a grid search with 5% of the dataset. Due to class imbalance where experts form the majority class, we use Synthetic Minority Oversampling Technique (SMOTE) (Chawla et al., 2002; Sharma et al., 2021b) to balance our training dataset. Such imbalances in data are not uncommon, where other fetal sonography studies have also had an imbalanced expert/beginner dataset (Sharma et al., 2021c; Wang et al., 2020). This imbalance is further amplified when considering separating sonographers on a per-year (of scanning experience) basis.

## 2.2. Predicting Skill Level of Fully Qualified Sonographers

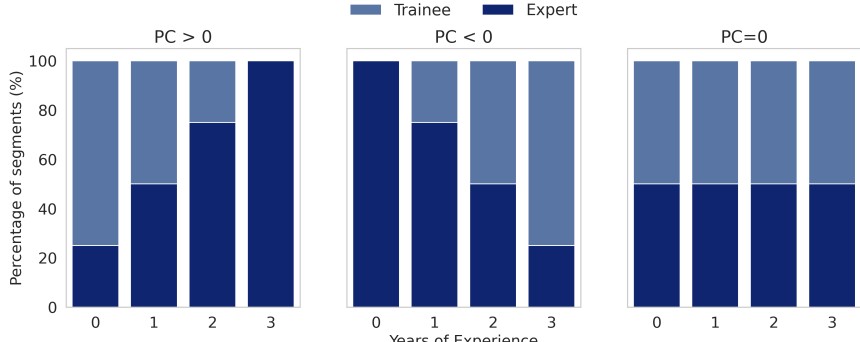

Figure 1: Example bar chart. Percentage of segments predicted as expert (dark blue) or trainee (light blue) grouped by the number of years of experience. *PC* refers to Pearson's correlation coefficient. *PC > 0* suggests that the years of experience and percentage of expert segments are positively correlated. *PC < 0* suggests that the years of experience and percentage of expert segments are negatively correlated. *PC = 0* suggests that the years of experience and percentage of expert segments are neither positively or negatively correlated.

We predict labels, expert or trainee, for live B-mode segments of FQ sonographers with a range of years of scanning experience. Then we calculate the proportion of segments that were labelled as expert and trainee, grouped by years of scanning experience, to test the hypothesis that years of scanning is analogous to skill (Sharma et al., 2021c; Wang et al., 2020; Lous et al., 2021). The trained skill classification model can identify expert segments which are more similar to trainee segments (i.e., expert segments which are misclassified as trainee segments), and whether the proportion of misclassified segments is significantly correlated with the number of years of scanning experience.

$$PC_{XY} = \frac{cov(X,Y)}{\sigma_X \sigma_Y} \tag{2}$$

We test the significance (at the 5% level) of years of experience and percentage of expert segments using Pearson's correlation coefficient (PC) (Equation (2)). The variables X and Y in Equation (2) are years of experience and percentage of expert segments (between 0 and 100%) respectively. $\sigma$ refers to the standard deviation, and *cov* refers to the covariance. The null hypothesis being tested is that there is no significant correlation (PC=0, Figure 1) between years of experience and percentage of expert segments. Bar charts are used for visual inspection of the proportions of expert and trainee labels. An example is shown in Figure 1.

We also investigate how the gaze skill classification model performs at the diagnostic plane task level. This is done by predicting on labelled diagnostic plane live B-mode segments. In our work, we use the head circumference (HC), abdominal circumference (AC), and heart plane finding tasks, which have been used in Wang et al. (2022, 2020); Sharma et al. (2021b). Briefly, heart plane finding or detection is considered to be more difficult to search for because the heart is smaller in size compared to the head or abdomen and requires subtle hand movements to find the heart planes. Therefore on average, we expect that the more experienced a sonographer is (in years), the more likely their live B-mode segments would be predicted as expert when considering the heart plane.

## 3. Data

| | Trainee 1 | Trainee 2 | Trainee 3 | Trainee 4 | Teacher |
|---|---|---|---|---|---|
| Number of sessions | 6 | 6 | 1 | 1 | 14 |

Table 1: Number of unique teacher-trainee scan sessions.

| Years of experience | 0 | 1 | 2 | 3 | 5 | 6 | 7 | 8 | 10 | 11 | 14 | 15 | 16 |
|---|---|---|---|---|---|---|---|---|---|---|---|---|---|
| Number of sessions | 136 | 115 | 33 | 8 | 22 | 16 | 5 | 4 | 18 | 13 | 104 | 39 | 2 |

Table 2: Number of unique scan sessions performed by fully qualified sonographers.

The sonographer's eye gaze data was acquired as part of the PULSE[1] (ERC-2015-AdG-694581) project which received ethics committee approval. We focus specifically on second-trimester scans which were the most commonly conducted. There are two dataset partitions

---

[1]. https://eng.ox.ac.uk/pulse/

used. One partition had a teacher (FQ sonographer with 5 years of experience) training 4 different trainees (under independent scan sessions) (Tab. 1). The second partition had 13 FQ sonographers within 0-16 years of scanning experience (Tab. 2).

**Gaze Preprocessing**   Eye-tracking and pupillary data were collected using a Tobii Eye Tracker 4C which was sampled at 90 Hz. We follow the pupillary data preprocessing method outlined in Sharma et al. (2021b), and the eye-tracking data preprocessing method outlined by Teng et al. (2021). Briefly, we discard any pupil diameters <1.5mm and >9.0mm, and linearly interpolate any missing values. For gaze data, we discard any segments with >210ms of gaze data missing and linearly interpolate any other gaps.

### 3.1. Training Data

| Model Grouping | Expert's Data Abbreviation | Expertise (years) | $\approx$ Class Imbalance Ratio |
|---|---|---|---|
| Teacher VS trainee | Teacher | 5 | 3 |
| | Teacher+$FQ_{2,5}$ | 2-5 | 12 |
| FQ sonographers VS trainee | Teacher+$FQ_{0,16}$ | 0-16 | 18 |
| | $FQ_{0,16}$ | 0-16 | 14 |
| FQ sonographer VS trainee | $FQ_{1,2}$ | 1-2 | 23 |
| | $FQ_{2,3}$ | 2-3 | 8 |
| | $FQ_{0,3}$ | 0-3 | 17 |
| | $FQ_{10,11}$ | 10-11 | 3 |
| | $FQ_{14,15}$ | 14-15 | 15 |

Table 3: Table of groups of experts represented in the training dataset for skill classification, with their corresponding number of years of scanning experience. The table also includes a class imbalance ratio of the expert class and trainee segments available for training; the expert class is the majority class. The abbreviation for these experts are $FQ_{a,b}$, where $FQ$ stands for fully qualified, and $a, b$ represents the lower and upper bound of number of years of scanning experience.

In our work, we considered different FQ sonographers to represent experts in our training dataset for skill classification. These models were outlined in Section 2.1 as *Teacher VS trainee*, *FQ sonographers VS trainee* and *FQ sonographer VS trainee*. *Teacher VS trainee* models used the same sonographer where they taught (Teacher) and performed scans on their own ($FQ_{2,5}$). Due to data imbalance (Tab. 2), we use sonographers with the most (top 5) gaze data in Tab. 2 to represent our expert population for our *FQ sonographer VS trainee* models.

We set aside 20% of the FQ data in Tab. 2, abbreviated as $FQ_{0,16}$, for training and testing our skill classification model. The remaining 80% is used to predict skill level of FQ sonographers. Any anatomy-specific segments were labelled using optical character recognition.

## 4. Results

### 4.1. Skill Classification

Table 4 shows the results of the model's performance on the test set across the 5 folds. On average, both LightGBM and XGBoost outperform CatBoost. This is not unexpected since the number of continuous features in the dataset is more than the number of categorical features. Given that class imbalance favours the majority class (expert), it is not surprising that the performance of the expert class is much better than that of the trainee class, with average F1 scores of at least 94%. The best performing model based on the trainee class performance uses an XGBoost architecture and $FQ_{10,11}$ as the expert. It achieves an F1 score of 95% for the expert class and 88% for the trainee class.

| Model | LightGBM | | XGBoost | | CatBoost | |
|---|---|---|---|---|---|---|
| Data | Expert | Trainee | Expert | Trainee | Expert | Trainee |
| Teacher | **0.94±0.01** | **0.79±0.02** | 0.94±0.01 | 0.78±0.03 | 0.91±0.00 | 0.71±0.01 |
| Teacher+$FQ_{2,5}$ | 0.97±0.00 | 0.72±0.04 | 0.97±0.00 | 0.74±0.02 | 0.95±0.01 | 0.62±0.04 |
| Teacher+$FQ_{0,16}$ | 0.98±0.00 | 0.60±0.02 | 0.98±0.00 | 0.60±0.02 | 0.95±0.01 | 0.38±0.03 |
| $FQ_{0,16}$ | **0.98±0.00** | **0.70±0.03** | 0.98±0.00 | 0.66±0.04 | 0.96±0.00 | 0.50±0.01 |
| $FQ_{1,2}$ | 0.99±0.00 | 0.71±0.04 | 0.99±0.00 | 0.74±0.01 | 0.97±0.00 | 0.58±0.02 |
| $FQ_{2,3}$ | 0.98±0.00 | 0.71±0.03 | 0.97±0.00 | 0.65±0.03 | 0.96±0.00 | 0.54±0.01 |
| $FQ_{0,3}$ | 0.99±0.00 | 0.84±0.02 | 0.99±0.00 | 0.80±0.02 | 0.98±0.00 | 0.68±0.04 |
| $FQ_{10,11}$ | 0.95±0.00 | 0.86±0.01 | **0.95±0.01** | **0.88±0.02** | 0.94±0.01 | 0.84±0.02 |
| $FQ_{14,15}$ | 0.98±0.00 | 0.72±0.02 | 0.98±0.00 | 0.71±0.02 | 0.95±0.01 | 0.52±0.02 |

Table 4: Average F1 scores using the different training datasets described in Tab. 3. In bold, the best performing model *Teacher VS trainee* model using LightGBM, the best performing model *FQ sonographers VS trainee* model using LightGBM, the best performing *FQ sonographer VS trainee* model using XGBoost.

*Teacher VS trainee*: A comparison of Teacher and Teacher+$FQ_{2,5}$ shows a 7% decrease in performance, suggesting that an expert's gaze is affected by the presence of a trainee. It could be that the expert teaches trainees in a specific textbook manner, but uses their own style when scanning individually.

*FQ sonographers VS trainee*: The best performing model which considers a range of years of experience achieves a 98% and 70% F1 score for expert and trainee classes respectively. This model only included sonographers performing scans individually. By including gaze data where the teacher was actively training a trainee, Teacher+$FQ_{0,16}$, the performance drops by 10%.

*FQ sonographer VS trainee*: The performance of the trainee class depends on which experts were used in training, with F1 scores between 71% and 86% (Table 4, LightGBM). When comparing similar years of experience, $FQ_{14,15}$ and $FQ_{10,11}$, $FQ_{0,3}$ and $FQ_{1,2}$, there is a difference of at least 13% (Table 4). These results suggest that when considering a skill classification model, a sonographer's style is also a factor that is not easily disentangled from their skill. As a result, misclassification of trainee segments is dependent on the style of the expert's gaze.

### 4.2. Skill Prediction of Fully Qualified Sonographers

We calculate the proportion of expert segments predicted by the models on 80% of $FQ_{0,16}$ and use bar charts to display the average proportion of segments predicted as trainee and expert. We use the LightGBM model as overall it had the best results (Table 4). For brevity, we show the bar charts of the top 3 sonographer-specific datasets which returned the best performing models: Teacher, $FQ_{10,11}$, $FQ_{0,3}$.

Figure 2(a) shows the proportion of expert-trainee labels, where the expert used in the training dataset was the Teacher with 5 years of experience. Note that sonographers with 2 and 5 years of experience have the highest proportion of their segments labelled as expert. This is because the Teacher also performed some scan sessions individually, which was presented in the $FQ_{0,16}$ dataset. In the $FQ_{0,16}$ dataset, the sonographer had 2 years of experience. At the time of teaching, they had 5 years of experience.

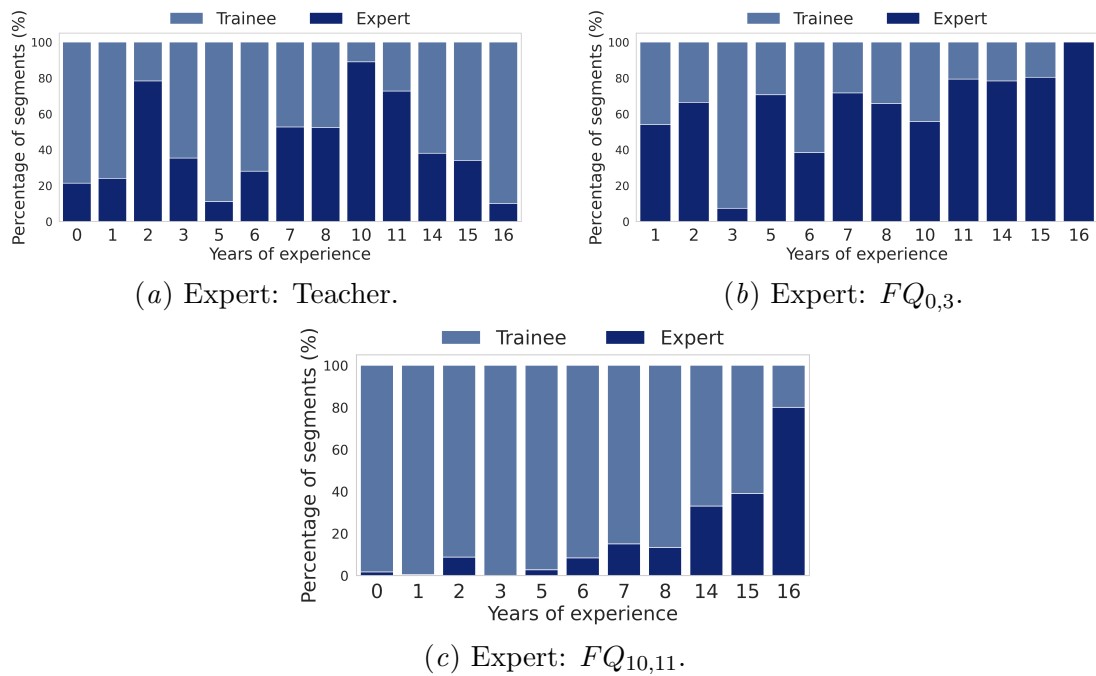

$(a)$ Expert: Teacher.

$(b)$ Expert: $FQ_{0,3}$.

$(c)$ Expert: $FQ_{10,11}$.

Figure 2: Percentage of segments predicted as expert (dark blue) or trainee (light blue) on $FQ_{0,16}$, grouped by the number of years of experience. The experts used for training the model are: Teacher, $FQ_{0,3}$ and $FQ_{10,11}$.

Both Figure 2(b) and Figure 2(c) show that the number of years of scanning is positively correlated with the proportion of segments being labelled as experts. The PC is significant (p-value < 0.05). This is unlike Figure 2(a) where the coefficient is -0.07 and is not significant.

| | Teacher | Teacher+$FQ_{0,16}$ | Teacher+$FQ_{2,5}$ | $FQ_{0,16}$ |
|---|---|---|---|---|
| p-value | 0.82 | 0.41 | 0.95 | 0.44 |
| PC | -0.07 | -0.26 | -0.02 | 0.25 |

| | $FQ_{1,2}$ | $FQ_{2,3}$ | $FQ_{0,3}$ | $FQ_{10,11}$ | $FQ_{14,15}$ |
|---|---|---|---|---|---|
| p-value | 0.15 | 0.46 | 0.00 | 0.00 | 0.0 |
| PC | -0.47 | -0.24 | 0.84 | 0.92 | 0.80 |

Table 5: Table of Pearson's coefficient (PC) and p-values between years of experience and percentage of expert segments predicted.

### 4.3. Anatomy Specific Skill

We also investigate how well our task-agnostic gaze skill classification model performs when considering specific anatomical planes. The anatomical planes that were considered are head circumference (HC), abdominal circumference (AC), and heart. The anatomy-specific PC results are shown in Table 6 which suggests that there is little significant correlation for anatomical planes AC and HC (2 out of 9 models show a significant correlation). Conversely, there is some significant correlation for heart planes (4 out of 9). These results suggest that there is significant difference in gaze behaviour between FQ sonographers when searching for the heart, but not for the HC and AC. The results of Sharma et al. (2021c,b) show that the heart is more difficult to search for because of its relatively smaller size in comparison to the abdomen and brain. Therefore, it is not surprising that the number of years of experience in scanning is positively correlated with the proportion of expert segments for the heart (Figure 3($a$) and Figure 3($b$)).

| | Teacher+$FQ_{2,3}$ | $FQ_{1,2}$ | $FQ_{0,3}$ | $FQ_{10,11}$ | $FQ_{14,15}$ |
|---|---|---|---|---|---|
| AC | 0.58 | | | 0.82 | |
| HC | 0.62 | | | 0.78 | |
| Heart | | -0.61 | 0.74 | 0.86 | 0.58 |

Table 6: Table of Pearson's coefficient (PC) between years of experience and percentage of expert segments predicted for head circumference (HC), abdominal circumference (AC), and heart. We only show the correlation coefficients which were found to be significant at the 5% level. Teacher, Teacher+$FQ_{0,16}$, $FQ_{0,16}$ and $FQ_{2,3}$ did not have any significant coefficients. Empty entries correspond to p-value > 0.05. Entries with a coefficient value correspond to p-values < 0.05.

Furthermore, we show the bar charts for $FQ_{0,3}$ and $FQ_{10,11}$. Visually, they confirm the results of the significance test, where there is a noticeable increase in predicted expert segments with years of experience for $FQ_{10,11}$ compared to $FQ_{0,3}$ for AC and HC.

A comparison between Figure 3($a$) and 3($b$) also show that there is a larger proportion of predicted expert segments when being compared against FQ sonographer with less scanning experience, $FQ_{0,3}$ (0-3 years), than that of $FQ_{10,11}$ (10-11 years). The same behaviour can

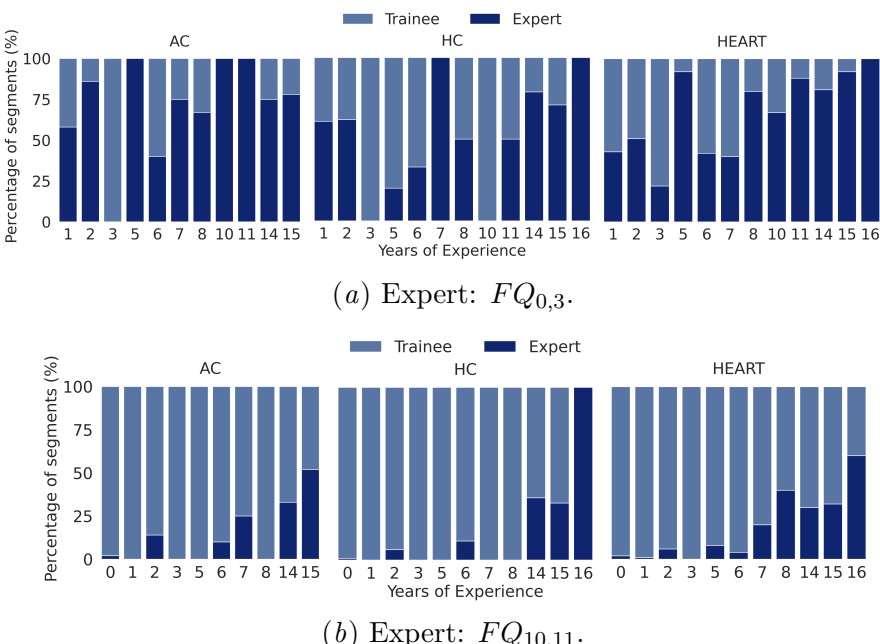

($a$) Expert: $FQ_{0,3}$.

($b$) Expert: $FQ_{10,11}$.

Figure 3: Percentage of segments predicted as expert (dark blue) or trainee (light blue) for anatomical planes HC, AC and Heart in $FQ_{0,16}$, grouped by number of years of experience. The experts used for training the model are: $FQ_{0,3}$ and $FQ_{10,11}$.

also be observed in Figure $2(b)$ and Figure $2(c)$. These results suggest that it is 'more difficult' to have gaze patterns of a sonographer who has been scanning for several years. It is likely that sonographers developed their own style over time, possibly moving away from their scanning style from earlier years when they first qualified.

## 5. Discussion

In our work, we have shown that the performance of a gaze skill classification model is dependent on the sonographer representing the expert population. We then used the model to predict whether a FQ sonographer's years of experience is positively correlated to the proportion of expert segments predicted. The Pearson's correlation coefficient test showed that when using $FQ_{0,3}$, $FQ_{10,11}$, and $FQ_{14,15}$ as the expert benchmark, there is a significant positive correlation between the number of scanning years and the percentage of expert segments. These results suggest that, without making any prior assumptions about the relationship between scanning years and expertise, there is a positive correlation between the 2 variables. With more years of scanning experience, a FQ sonographer is likely to have a higher proportion of predicted expert segments. This relationship is also seen when sonographers are searching for heart planes.

The trainee class was highly imbalanced in some of the training data, such as $FQ_{0,3}$ and $FQ_{1,2}$. A comparison of $FQ_{0,3}$ and $FQ_{10,11}$, which had an imbalance ratio of 17 and 3

respectively, returned similar results for the best-performing model. When comparing $FQ_{0,3}$ and $FQ_{1,2}$, both had between 0-3 years of experience but a 13% difference in performance for the trainee class. Similarly, $FQ_{10,11}$ and $FQ_{14,15}$ had a 14% difference. These results suggest that although class imbalance could have caused the minority class (trainee) to perform worse than the expert class, it is more likely that the gaze behaviour of a sonographer is dependent on their scanning style, causing different representations of experts to return a range of model performances.

Some considerations of the dataset which are important to note are as follows. The PULSE data was collected from a single site and used the same ultrasound scanning machine. We also only consider second trimester scans in our work. The fetus in the first and third trimesters would present differently during the scan, and it would be useful to see whether our method can generalise across different trimesters, and between different ultrasound machines.

## 6. Conclusion

In this paper, we have presented a skill classification model, where experts were defined as fully qualified sonographers independent of their years of scanning experience, and trainees were defined as sonographers learning how to scan. Our best performing model considering a range of years of experience used a LightGBM and returned F1 scores of 98% and 70% for expert and trainee classes respectively. We have also showed that sonographer gaze behaviour is indicative of both skill and style, with performance differences of up to 16%. Finally, without making any prior assumptions of the correlation between years of experience as a direct measure of skill, we show that there is a significant positive correlation between years of scanning and expertise when considering task-agnostic gaze characteristics and task-specific planes such as the heart.

### Acknowledgments

We thank Qianhui Men and Mohammad Alsharid for proof-reading the paper. We acknowledge the ERC (Project PULSE: ERC-ADG-2015 694581). ATP is supported by the Oxford Partnership Comprehensive Biomedical Research Centre with funding from the NIHR Biomedical Research Centre (BRC) funding scheme. This work was also supported in part by the InnoHK-funded Hong Kong Centre for Cerebro-cardiovascular Health Engineering (COCHE) Project 2.1 (Cardiovascular risks in early life and fetal echocardiography).

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
