# OpenReview forum: "Skill, or Style? Classification of Fetal Sonography Eye-Tracking Data"
_NeurIPS.cc/2022/Workshop/GMML — Gaze Meets ML 2022 Poster_

### Official Review · Reviewer_FsmQ · 2022-10-14
**The study of the correlation between years of experience and expertise levels is interesting, but the clarity of the paper and the experimental design need to be improved.**

**Rating:** 6
**Confidence:** 4

**Review:**

This paper presents a study of using the eye-tracking and pupillary data of fetal ultrasound sonographers to classify between experts and trainees. The correlation between years of scanning experience and the proportion of predicted expert labels of fully qualified sonographers was also studied.

The introduction is well-written, but the key idea of using the predicted expert segments is not well-explained.

1. The most confusing part is the use of the predictions from a trained model to study the correlation between years of experience and expertise levels. Can such correlation be obtained using the ground truth? The answer is probably negative if all segments from experts are treated as expert segments, as the “percentage of expert segments” is 100% and the correlation coefficient cannot be computed with sigma_y = 0 in eq. (2).  The same thing happens if the trained model can perfectly classify all segments of experts as expert segments. Therefore, eq. (2) is only valid when the trained model misclassifies expert segments as trainee segments. If this is intentional, please explain. For example, the trained model can help to identify the segments of experts that are similar to those of trainees, even though they are marked as expert segments in the ground truth.

2. The size of the dataset is small especially for some years of experience, for example, the numbers of sessions with 3, 7, 8, and 16 years of experience were less than 10. This makes the corresponding results less reliable. The data can be combined into different groups so that each group can have a decent number of samples.

3. What does the “3” in “3 × n” features represent at line 115?

4. At line 171, “The second partition had 13 FQ sonographers within 0-13 years of scanning experience (Tab. 2)”. Table 2 shows up to 16 years of experience.

---

### Official Review · Reviewer_nJ5k · 2022-10-17
**A well written and novel contribution**

**Rating:** 8
**Confidence:** 4

**Review:**

* The authors explore the relationship between human skill at fetal ultrasound scanning with gaze information, and strong positive correlations through learnt models.
* A well written and clear structured contribution with discussions and results supporting the findings uncovered through the experiments.
* Discussions regarding the impact of the features and processing used on the predictor performance would be interesting to see.
* A qualitative understanding of any gaze patterns which are correlated with experts would be very insightful and could aid research into effective methods and aids for trainees.
* Overall, a very well written paper with interesting findings and contributions which should be shared with the broader community.

---

### Meta-Review · Area_Chair_Xuj6 · 2022-10-20

**Recommendation:** Accept (Poster)
**Confidence:** 4

**Metareview:**

This work investigates the relationship between sonographers at different skill levels at a fetal ultrasound scanning task, based on gaze information and eye movements when performing the task. Reviewers have indicated that the paper presentation is clear and the motivation is adequately detailed. Some concerns were raised about the dataset size and the use of expert segments. The authors can consider adding more discussion to the camera-ready to either resolve some of the reviewer questions or describe the limitations left to future work. Overall recommend acceptance based on the positive feedback and the fit to the target audience.

---

### Decision · Program_Chairs · 2022-10-20

Accept (Poster)